# Risk Factors on the Progression to Clinical Outcomes of COVID-19 Patients in South Korea: Using National Data

**DOI:** 10.3390/ijerph17238847

**Published:** 2020-11-28

**Authors:** Seon-Rye Kim, Seoul-Hee Nam, Yu-Rin Kim

**Affiliations:** 1Department of Pharmacy, College of Pharmacy, Kangwon National University, Chuncheon-si 24341, Korea; sjsanj@hanmail.net; 2Department of Dental Hygiene, College of Health Science, Kangwon National University, 346 Hwangjo-gil, Dogye-up, Samcheok-si, Gangwon-do 25945, Korea; 3Department of Dental Hygiene, Silla University, 140 Baegyang-daero, 700 beon-gil, Sasang-gu, Busan 46958, Korea

**Keywords:** COVID-19, risk factor, severity, ICU, blood variables

## Abstract

10–20% of COVID (Corona Virus Disease)-19 cases proceed to a severe stage, and age and the presence of comorbidity increased the risk of death from COVID-19. The identification of risk factors on progression to the severity stages is essential in providing more efficient and suitable management to COVID-19 patients. However, there is insufficient study on risk factors for severity stages of COVID-19 patients. In this study, 2959 confirmed COVID-19 patients were analyzed while using national data, COVID-19 patients Clinical Epidemiological Information provided from the Korea Disease Control and Prevention Agency. The epidemiological variable, hospital room, periods from confirmation to release, initial symptom and vital signs, underlying comorbidities, and initial blood variables were used to verify the relation with progression to severity stages of COVID-19 and severe COVID-19. The chi-square test, welch test, multiple regression and logistic regression analysis were performed. The ICU (Intensive Care Unit) admission rate of patients having characteristics, such as older age, male, abnormal BMI (Body Mass Index), high heart rate, high body temperature, fever, cough, sputum, sore throat, rhinorrhea, fatigue, dyspnea, change of consciousness, diabetes mellitus, hypertension, chronic artery disease, chronic kidney disease, cancer, dementia, abnormal hemoglobin, abnormal hematocrit, abnormal lymphocyte, abnormal platelets, and abnormal white blood cell were high. The risk factors for severe COVID-19 were older age, shorter hospitalization, abnormal lymphocyte, abnormal platelets, dyspnea, change of consciousness, and dementia. Whereas, significant predictors for progression to severity stages of COVID-19 were older age, longer period from confirmation to release, higher BMI, higher body temperature, abnormal lymphocyte, abnormal platelets, fever, no sore throat, dyspnea, no headache, COPD (Chronic Obstructive Pulmonary Disease), and dementia. Therefore, classifying patients with a high risk of severe stage of COVID-19 and managing patients by considering the risk factors could be helpful in the efficient management of COVID-19 patients.

## 1. Introduction

Coronavirus has a positive-sense single-stranded RNA genome, and its helical symmetry nucleocapsid is approximately 26–32 kb in size [1,2]. It is highly infectious, and the clinical symptoms of COVID-19, the disease that is caused by it, include fever, dry cough, myalgia, and fatigue, with the severe cases progressing to acute respiratory distress syndrome, leading to bleeding and coagulation dysfunction [3,4].

After December 2019, COVID-19 has rapidly out-broken in 214 countries to date. Because of the rapid increasing of infected patients, the World Health Organization finally declared a pandemic, as COVID-19 is a pandemic when it spreads over an entire country or globally [5,6]. South Korea confirmed its first COVID-19 case in late January 2020. In February, the country experienced an exponential spike in the number of COVID-19 cases caused by a “super-spreader” [7]. After that time, in South Korea, the COVID-19 infection response guidelines were strengthened and the infectious-disease crisis warning was raised to “severe” on February. Additionally, ‘the COVID-19 Code of Conduct’ to be observed at the severe disease stage was announced, and strong “social distancing” was implemented from March [8].

As of 12 September 2020, a total of 29,191,113 people were confirmed and 933,843 people were dead in the world. The mortality rate for COVID-19 is around 3.2%, which is higher than that of Spanish influenza (2.5%) [6,9,10]. Approximately 80% of COVID-19 patients have mild conditions; but around 20% of COVID-19 patients have progressed to severe stage [11]. Scientists all over the world are developing efficient medicines and vaccines for COVID-19 disease, but it will take a long time for it to be commercialized. Therefore, identifying the characteristics and risk factors of COVID-19 could provide valuable lessons.

Zhou et al. reported that potential risk factors were older age and high SOFA score in 191 patients of China [11]. Additionally, there is a study that the deaths in China and Italy are similar in mostly the elderly with comorbidities [12]. Yang et al. found that underlying disease, including hypertension, respiratory disease, and cardiovascular disease, elevated the risk for severe patients [13]. Emami et al. summarized previous researches and mentioned that the risk factors for mortality of COVID-19 included comorbidities, such as hypertension, coronary disease, and diabetes mellitus, as well as abnormalities, including cardiac troponin, interleukin-6, and C-reactive protein [14]. Like these, most studies have reported risk factors of mortality or severe stage in COVID-19 patients, as older age and comorbidity increased risk of death from COVID-19 [12,13,14].

Nevertheless, there is lack of research regarding the risk factors that are related to progression to the severity stages of COVID-19. The identification of risk factors on progression to the severity stages is essential in providing more efficient and suitable management to COVID-19 patients. Our study began to answer important questions on COVID-19 progression and outcomes, as well as potential risk factors bringing to intensive care unit admission. We analyzed risk factors on the progression to severity stages of COVID-19 while using national data that were provided by the Ministry of Health and Welfare. This study enrolled all patients who were released from isolation or dead after the confirmation of COVID-19 in South Korea until 30 April 2020.

## 2. Materials and Methods 

### 2.1. Study Subjects

This retrospective cohort study investigated all patients from COVID-19 patients Clinical Epidemiological Information provided from Korea Disease Control and Prevention Agency (http://www.kdca.go.kr/). Recently, Korea Disease Control and Prevention Agency agreed to the share valuable national data that were related to COVID-19 patients for public health purposes. This data included epidemiological variable, initial symptom and vital signs, underlying comorbidities, initial blood variables, hospital room, periods from confirmation to release, and the severity stage of COVID-19. As of 30 April 2020, patients who were confirmed to have been released from isolation or dead among COVID-19 confirmed patients were targeted. Original data included 5628 patients. However, 2959 patients were finally analyzed, excluding data with pregnancy-related variables or missing values for other variables.

The cured group included COVID-19 patients who were released from isolation after treatment by 30 April 2020. The dead group included COVID-19 patients that were dead after COVID-19 confirmation by 30 April 2020. This study was approved by Institutional Review Board of Silla University (No. 104149-20207-HR-03), and it was conducted in compliance with the Helsinki Declaration.

### 2.2. Data Collection

We extracted epidemiological, clinical, and outcome data from COVID-19 patients Clinical Epidemiological Information provided from Korea Disease Control and Prevention Agency. The severity stages of COVID-19 were classified into eight levels. such as no disruption to daily life (2291 patients), hindrance to daily life and no oxygen required (152 patients), oxygen treatment via nasal cannula (321 patients), oxygen mask (26 patients), non-invasive ventilation (26 patients), invasive ventilation (16 patients), multi-organ damage and extracorporeal membrane oxygenation (ECMO) (nine patients), and death (118 patients). The severe COVID-19 were classified into two groups, such as severe case and non-severe case. A severe case included no disruption to daily life, hindrance to daily life and no oxygen required, oxygen treatment via nasal cannula, and oxygen mask. A non-severe case included non-invasive ventilation, invasive ventilation, ECMO, and death. Age and gender were investigated. The days of isolation was investigated as a continuous variable. The initial symptoms and vital signs were investigated. The body mass index (BMI), systolic blood pressure (SBP), and diastolic blood pressure (DBP) were investigated as continuous variables. In this study, they were divided into a normal group and abnormal group. Additionally, heart rate and body temperature were investigated as continuous variables. Clinical findings include fever (≥37.5 °C), cough, sputum, sore throat, rhinorrhea, myalgia, fatigue, dyspnea, headache, change of consciousness, vomiting/nausea, diarrhea, diabetes mellitus (DM), hypertension, heart failure, chronic cardiovascular disease, asthma, chronic obstructive pulmonary disease (COPD), chronic kidney disease (CKD), cancer, rheumatoid/auto-immune disease, and dementia. Hemoglobin (Hgb), hematocrit (HCT), lymphocyte, platelets (PLT), and white blood cell (WBC) were investigated as continuous variables. In this study, according to the normal range of each variable, we analyzed by dividing into normal group and abnormal group.

### 2.3. Statistical Analyses

The independent *t*-test, Welch test, Cochran test, and chi-square tests were performed in order to verify the differences in epidemiological and clinical data between general ward (GW) group and intensive care unit (ICU) group. Additionally, we analyzed multivariate logistic regression to analyze the correlations between epidemiological and clinical variables and non-severe/severe COVID-19. The multivariate regression analysis through the stepwise backward performed the correlations between the epidemiological and clinical variables and the severity of COVID-19. All of the statistical analyses were performed while using R 3.4.1 for Windows & RStudio1.0.136_Windows Vista/7/8/9 (RStudio, PBC. All Rights Reserved, 250 Northern Ave, Boston, MA 02210, USA). We used the significance threshold of *p* < 0.05, and all tests were two-tailed.

## 3. Results

### 3.1. Epidemiological and Clinical Caracteristics by Hospital Room

The final analysis in this study included 2959 patients who were confirmed to have been released from isolation among patients with COVID-19 confirmed cases. Of the 2959 patients analyzed in Korea, 133 patients were admitted in ICU. The ICU admission rate of male was 60.9%, higher than that of female (39.1%). Older age was also associated with a higher ICU admission rate. Death was also associated with a higher ICU admission rate (42.1%). The ICU admission rate of patients with abnormal BMI (71.4%) was approximately 2.3 times higher than that of patients with normal BMI (28.6%). The median heart rate in ICU (88.84 n/min.) was significantly higher than that in GW (85.08 n/min). The median body temperature in ICU (37.3 °C) was significantly higher than that in GW (36.9 °C). The initial symptoms, including fever, cough, sputum, sore throat, rhinorrhea, fatigue, dyspnea and change of consciousness, were associated with ICU admission. Hypertension, DM, coronary artery disease, chronic kidney disease, cancer, and dementia were associated with ICU admission, and the ICU admission rate of patients with those underlying diseases was high. For variables that were related to blood test in initial time, significant differences in ICU admission rate were observed in all variables. The ICU admission rate of patients who have abnormal hemoglobin, abnormal hematocrit, abnormal lymphocyte, abnormal platelets, and abnormal WBC, was 36.8%, 40.6%, 65.4%, 27.8%, and 31.6%, respectively. The patients with abnormal blood variables had a higher ICU admission rate, whereas those with normal blood variables had a lower ICU admission rate (Table 1). 

### 3.2. Multiple Logistic Regression for Risk Factors

The results of logistic regression analysis for severe COVID-19 patients are as follows (Figure 1): in terms of epidemiological characteristics, every 10 years increase (OR: 1.88, 95% CI: 1.44–2.46) and period increase from confirmation to release (OR: 0.97, 95% CI: 0.95–0.99) were associated with a higher risk of severe COVID-19. For initial symptoms, dyspnea (OR: 6.51, 95% CI: 3.39–12.51) and change of consciousness (OR: 29.33, 95% CI: 2.82–304.79) increased the risk of severe COVID-19. There was a higher risk of being severe COVID-19 patients with abnormal lymphocyte (OR: 5.57, 95% CI: 2.87–10.82), abnormal platelets (OR: 3.01, 95% CI: 1.52–5.99), and dementia (OR: 3.51, 95% CI: 1.33–9.29).

### 3.3. Multiple Regression for Risk on Severity of COVID-19

Stepwise backward multiple linear regression analysis built a reasonable model, F (43, 2915) = 222.4, *p* < 0.001, explaining 76.3% of the variance in the progression to severity stages of COVID-19 (R^2^ = 0.763). We analyzed multiple linear regression, except hemoglobin and hematocrit, because of multicolinearity between factors. No violations of linearity were detected. The results of multiple linear regression analysis on the progression to severity stages of COVID-19 are as follows. In terms of epidemiological characteristics, every 10 year increase (0.069 times), one day longer hospitalization (0.008 times), 1 BMI increase (0.039 times), and 1 °C increase (0.110 times) were related with risk on the progression to severity stages of COVID-19. For initial symptoms, fever (0.176 times), sore throat (−0.113 times), dyspnea (0.620 times), and headache (−0.084 times) increased the risk on the progression to severity stages of COVID-19. There was a higher risk of deteriorating COVID-19 patients with abnormal lymphocyte (0.317 times), abnormal platelets (0.227 times), COPD (0.302 times), and dementia (0.309 times). Significant predictors for the progression to severity stages of COVID-19 were older age, longer hospitalization, higher BMI, higher body temperature, abnormal lymphocyte, abnormal platelets, fever, no sore throat, dyspnea, no headache, COPD, and dementia (Table 2).

## 4. Discussion

This study used data of COVID-19 patients’ Clinical Epidemiological Information provided from the Korea Disease Control and Prevention Agency. The analysis included 2959 patients who had complete values of all variables, except pregnancy. Using these data, we found that risk factors for progression to the severity stages of COVID-19 and risk factors for severe COVID-19 were not same. The risk factors for severe COVID-19 were older age, shorter period from confirmation to release, abnormal lymphocyte, abnormal platelets, dyspnea, change of consciousness, and dementia. The risk factors for the progression to severity stages of COVID-19 were older age, longer period from confirmation to release, higher BMI, higher body temperature, abnormal lymphocyte, abnormal platelets, fever, no sore throat, dyspnea, no headache, COPD, and dementia. Consequently, the risk factors for progression to the severity stages of COVID-19 were more than those for severe COVID-19.

As we know, this study is the first try analyzing risk factors related to the severity of COVID-19 according to progressive stages. Hence, the meaningful point was that the main outcomes of this study were not risk factors for death, but risk factors for progression to the severity stages of disease. Among patients with older age, the OR for severity of COVID-19 was about two times higher than that of 10 younger aged COVID-19 patients, which was statistically significant in our study, as already known. Previous studies reported age as the most significant predictor of death or severe stage in COVID-19 patients [12,13,14,15,16,17,18,19,20,21,22,23,24,25,26,27,28,29,30,31,32]. This study revealed that older age was a risk factor for progression to the severity stage as well as severe COVID-19. The older age could affect immune systems, which destroys viral replication [19,20,21,22,23,24]. If age was considered to be a modifier, the result could be different. However, this study focused on characteristics in a clinical situation, so the result revealed that older age increased the risk for progression to the severity stage. A longer period from confirmation to release increased the risk on the progression to severity stage of COVID-19, but a shorter period from confirmation to release increased risk on severe COVID-19 in this study. That result might be because of rapid deterioration of disease. This is the same as Wang et al.’s study, which was analyzed according to severity [32]. Their study is consistent with the results of a short-term death, with the number of white blood cells increasing just a week after the symptoms began, which results in a decrease in the number of lymphocytes. Additionally, this result was consistent with a recent study reported that a shorter length of stay at the time of first admission was related to higher rate of readmission [15]. Higher BMI and higher body temperature increased the risk of progression to the severity stages of COVID-19. This result is consistent with previous researches regarding severe COVID-19 [15,25,27,28]. Especially, Chang et al. reported that body temperature, chills, and DM were associated with the aggravation of COVID-19 [26]. However, this study showed different results that higher BMI and higher body temperature increased the risk for progression to the severity stages of COVID-19, whereas higher BMI and higher body temperature did not increase risk for severe stage COVID-19. Abnormal lymphocyte and abnormal platelets increased the risk of progression to the severity stages of COVID-19. Lymphocytes are important to immunological response, such as cytokines and chemokines [16,17]. Other studies mentioned that patients with abnormal lymphocyte and abnormal platelets were vulnerable to immune system, which could deteriorate general condition and worsen COVID-19 symptoms [16,17,18]. Therefore, abnormal lymphocyte and abnormal platelets were risk factors for severe COVID-19 as well as the progression to severity stages of COVID-19. The sore throat and headache decreased risk on progression to the severity stages of COVID-19. Although the mechanisms of these associations were not clear, the protective effect might be a result of medications taken to treat these symptoms. In terms of initial symptoms, dyspnea and change of consciousness were significantly associated with an increased risk of progression to the severity stage COVID-19. Dyspnea is a representative symptom of respiratory disease; therefore, dyspnea can be an important risk factor for progression to the severity stage COVID-19 [10,11,12,13,14,28]. Therefore, dyspnea was risk factor for severe COVID-19 as well as the progression to severe stages of COVID-19.

In this study, COPD and dementia were associated with increased risks of severity of COVID-19. Nevertheless, only dementia was associated with an increased risk of the severe stage COVID-19. COPD is a representative respiratory disease; furthermore, COPD can worsen the severity of COVID-19 [19,20,21,22]. In terms of dementia, when interpreting this result, it is needed to consider the epidemiological characteristics. Specifically, dementia might be related with the large clustered outbreak in closed psychological hospital and nursing home. Additionally, early reports of COVID-19 outbreak mentioned that the accompanying low socioeconomic level could increase the risk of COVID-19, rather than psychological disorder [18,19,20] In addition, it was reported that comorbidities, including DM, hypertension, and chronic respiratory disease, except for cancer, were related to death in early studies [19,20,21,22,23,24,25,26,27,28,29,30,31,32,33]. However, this study showed different results that COPD and dementia increased the risk for progression to the severity stages of COVID-19, whereas only dementia increased the risk for severe stage COVID-19.

Our study found more risk factors, such as older age, longer period from confirmation to release, higher BMI, higher body temperature, abnormal lymphocyte, abnormal platelets, fever, no sore throat, dyspnea, no headache, COPD, and dementia for the progression to severity stages COVID-19 disease. Furthermore, as this study used all patients of South Korea and patients with various progression stages of COVID-19 disease, the result could be very useful for the management of COVID-19 patients. The limitation of the study is that the national data that were provided by the Korea Centers for Disease Control and Prevention (KCDC) in Korea were too limited for the protection of personal information of Corona 19. In addition, the analysis method and the duration of the analysis were limited, so the analysis was not freely conducted. Above all, in spite of all patients of COVID-19 as of 30 April 2020, in Korea, this study included relatively insufficient numbers of severe stage COVID-19 patients. Therefore, more studies needed to be conducted with larger patient numbers.

## 5. Conclusions

Approximately 20% of COVID-19 patients progressed to the severe stage. We found that older age, longer period from confirmation to release, higher BMI, higher body temperature, abnormal lymphocyte, abnormal platelets, fever, no sore throat, dyspnea, no headache, COPD, and dementia significantly increased risk for the progression to severity stages of COVID-19. Therefore, we suggest that clinicians should consider these risk factors on progression to the severity stages for more efficient management of COVID-19 patients.

## Figures and Tables

**Figure 1 ijerph-17-08847-f001:**
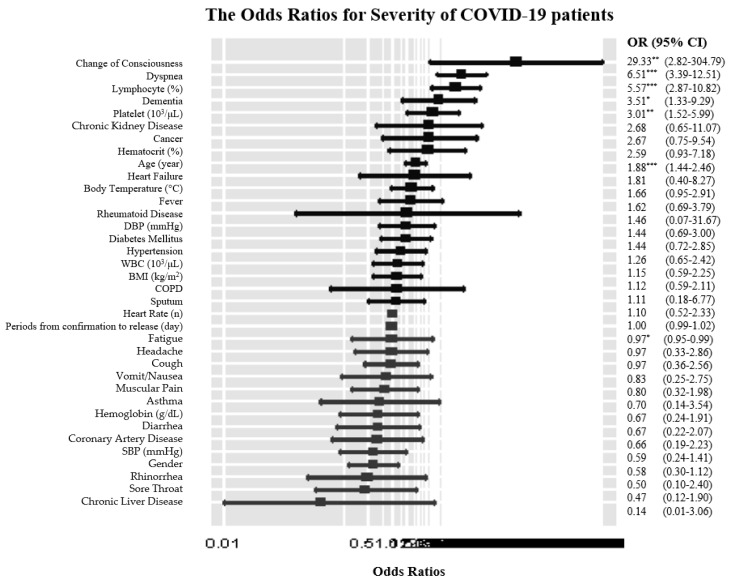
The Odds Ratios for Severe COVID-19 patients. * (*p* < 0.05), ** (*p* < 0.01), *** (*p* < 0.001), BMI (Body Mass Index), SBP (Systolic Blood Pressure), DBP (Diastolic Blood Pressure), COPD (Chronic Obstructive Pulmonary Disease, OR (odds ratio), CI (confidence interval), WBC (white blood cell).

**Table 1 ijerph-17-08847-t001:** Epidemiological and Clinical Variables by hospital room.

Variables	Sub-Items	GW (*n* = 2826)	ICU (*n* = 133)	*p*
^¥^ Age (year) *n* (%)	0–9	28 (100)	0(0)	0.000 ***
	11–10	112 (100)	0(0)	
	20–29	485 (99.0)	5(1.0)	
	30–39	277 (97.5)	7(2.5)	
	40–49	368 (99.2)	3(0.8)	
	50–59	601 (97.2)	17(2.8)	
	60–69	505 (93.7)	34(6.3)	
	70–79	276 (85.2)	48(14.8)	
	80-	174 (90.2)	19(9.8)	
Gender *n* (%)	Male	1098 (38.9)	81 (60.9)	0.000 ***
	Female	1728 (61.1)	52 (39.1)	
Release or Death *n* (%)	Death	62 (2.2)	56 (42.1)	0.000 ***
	Release	2764 (97.8)	77 (57.9)	
BMI (kg/m^2^) *n* (%)	Normal	1178 (41.7)	38 (28.6)	0.004 **
SBP (mmHg) *n* (%)	Normal	687 (24.3)	26 (19.5)	0.25
DBP (mmHg) *n* (%)	Normal	1049 (37.1)	61 (45.9)	0.052
^†^ Heart Rate (*n*) M ± SD		85.08 ± 14.73	88.84 ± 19.76	0.032 *
^†^ Body Temperature (°C) M ± SD		36.91 ± 0.56	37.31 ± 0.81	0.000 ***
^†^ Periods from confirmation to release (day) M ± SD		25.95 ± 11.21	26.78 ± 15.26	0.534
Fever *n* (%)	Yes	641 (22.7)	66 (49.6)	0.000 ***
Cough *n* (%)	Yes	1245 (44.1)	72 (54.1)	0.028 *
Sputum *n* (%)	Yes	824 (29.2)	51 (38.3)	0.030 *
Sore Throat *n* (%)	Yes	462 (16.3)	10 (7.5)	0.009 **
Rhinorrhea *n* (%)	Yes	259 (9.2)	9 (6.8)	0.043 *
Muscular Pain *n* (%)	Yes	432 (15.3)	21 (15.8)	0.973
Fatigue *n* (%)	Yes	141 (5.0)	13 (9.8)	0.026*
Dyspnea *n* (%)	Yes	331 (11.7)	70 (52.6)	0.000 ***
Headache *n* (%)	Yes	452 (16.0)	15 (11.3)	0.181
Change of Consciousness *n* (%)	Yes	7 (0.2)	7 (5.3)	0.000 ***
Vomit/Nausea *n* (%)	Yes	151 (5.3)	10 (7.5)	0.376
Diarrhea *n* (%)	Yes	216 (7.6)	15 (11.3)	0.173
Diabetes Mellitus *n* (%)	Yes	392 (13.9)	39 (29.3)	0.000 ***
Hypertension *n* (%)	Yes	633 (22.4)	64 (48.1)	0.000 ***
^¥^ Heart Failure *n* (%)	Yes	30 (1.1)	4 (3.0)	0.063
Coronary Artery Disease *n* (%)	Yes	102 (3.6)	10 (7.5)	0.038 *
Asthma *n* (%)	Yes	75 (2.7)	5 (3.8)	0.406
^¥^ COPD *n* (%)	Yes	26 (0.9)	2 (1.5)	0.361
Chronic Kidney Disease *n* (%)	Yes	27 (1.0)	10 (7.5)	0.000 ***
Cancer *n* (%)	Yes	86 (3.0)	9 (6.0)	0.033 *
^¥^ Chronic Liver Disease *n* (%)	Yes	44 (1.6)	2 (1.5)	1
^¥^ Rheumatoid Disease *n* (%)	Yes	24 (0.8)	1 (0.8)	1
Dementia *n* (%)	Yes	103 (3.6)	10 (7.5)	0.041 *
Hemoglobin(g/dL) *n* (%)	Normal	2347 (83.1)	84 (63.2)	0.000 ***
Hematocrit (%) *n* (%)	Normal	2267 (80.2)	79 (59.4)	0.000 ***
Lymphocyte (%) *n* (%)	Normal	2171 (76.8)	46 (34.6)	0.000 ***
Platelet (10^3^/μL) *n* (%)	Normal	2470 (87.4)	96 (72.2)	0.000 ***
WBC (10^3^/μL) *n* (%)	Normal	2223 (78.7)	91 (68.4)	0.007 **

By chi-square test, ^¥^ Fisher’s exact test (*n* < 5), ^†^ Welch’s test (Bartlett test of homogeneity of variances; *p* < 0.05), * (*p* < 0.05), ** (*p* < 0.01), *** (*p* < 0.001), BMI (Body Mass Index), SBP (Systolic Blood Pressure), DBP (Diastolic Blood Pressure), COPD (Chronic Obstructive Pulmonary Disease), GW (general ward), ICU (intensive care unit).

**Table 2 ijerph-17-08847-t002:** Multiple Regression for Risk on the Severity of COVID19.

Variables	Estimate	S.E.	T	*p*
Age	0.069	0.0096	7.273	0.000 ***
Periods from confirmation	0.008	0.0013	6.039	0.000 ***
to release (day)				
BMI (kg/m^2^)	0.039	0.0193	2.048	0.040 *
Body Temperature (°C)	0.11	0.0313	3.776	0.000 ***
Lymphocyte (abnormal)	0.317	0.0393	8.064	0.000 ***
Platelets (abnormal)	0.227	0.0464	4.886	0.000 ***
Fever	0.176	0.0423	4.174	0.000 ***
Sore Throat	−0.113	0.0396	−2.848	0.004 **
Dyspnea	0.62	0.0448	13.847	0.000 ***
Headache	−0.084	0.0399	−2.099	0.036 *
COPD	0.302	0.15	2.016	0.044 *
Dementia	0.309	0.0814	3.727	0.000 ***

* (*p* < 0.05), ** (*p* < 0.01), *** (*p* < 0.001), S.E. (Standard Error), BMI (Body Mass Index), SBP (Systolic Blood Pressure), DBP (Diastolic Blood Pressure), COPD (Chronic Obstructive Pulmonary Disease).

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
