# Peer review of "Risk Factors on the Progression to Clinical Outcomes of COVID-19 Patients in South Korea: Using National Data"

_ijerph, 2020, doi:10.3390/ijerph17238847_

Round 1

Reviewer 1 Report

This study is very important in exploring the risk factors related to progression to the severity stages of COVID-19 for the first time. The data that has been used is wide enough to provide a solid result.

The sentence “ICU admission rate of older age…” in the abstract is not clear. The abstract is misleading with the number of cases analyzed-since only 2,959 cases were finally analyzed.

Study subjects- There is a need to indicate how many of the cases were of people who died and how many of the cases released from isolation. In addition, it is not clear who is part of the sample- in the method: “…patients who were confirmed to have been released from isolation or dead” but in the results “The final analysis in this study included 2959 patients who were confirmed to have been released 122 from isolation”. The full and accurate data should appear in the method.

Figure 1- it should clearly state why some of the risk factors are in blue and others in red. The scale below the figure is unclear.

Discussion:

The sentence “that result might be because of rapid deterioration of disease.” should be further explained.

“Whereas, only dementia was associated with an increased risk of the severe stage COVID-19”- was associated by whom? The previous sentence already described the findings of the current study.

The conclusion should be written in more detail, including more recommendations for practice and research and the limitations of the study.

Author Response

Open Review 1

Comments and Suggestions for Authors

This study is very important in exploring the risk factors related to progression to the severity stages of COVID-19 for the first time. The data that has been used is wide enough to provide a solid result.

The sentence “ICU admission rate of older age…” in the abstract is not clear.

The abstract is misleading with the number of cases analyzed-since only 2,959 cases were finally analyzed.

à We have modified it to your opinion.

Study subjects- There is a need to indicate how many of the cases were of people who died and how many of the cases released from isolation.

à We have modified it to your opinion.

In addition, it is not clear who is part of the sample- in the method: “…patients who were confirmed to have been released from isolation or dead” but in the results “The final analysis in this study included 2959 patients who were confirmed to have been released 122 from isolation”. The full and accurate data should appear in the method.

à The full original data from KCDC included 5,628 patients who were confirmed to have been released from isolation or dead among COVID-19 confirmed patients. But 2,959 patients were finally analyzed, excluding data with pregnancy-related variables or missing values for other variables.

Figure 1- it should clearly state why some of the risk factors are in blue and others in red. The scale below the figure is unclear.

à We have modified it to your opinion.

Discussion:

The sentence “that result might be because of rapid deterioration of disease.” should be further explained.

à We have modified it to your opinion.

“Whereas, only dementia was associated with an increased risk of the severe stage COVID-19”- was associated by whom? The previous sentence already described the findings of the current study.

à The results of logistic regression analysis for severe COVID-19 patients showed that only dementia was associated with an increased risk of the severe stage COVID-19.

The conclusion should be written in more detail, including more recommendations for practice and research and the limitations of the study

à We have modified it to your opinion.

We are deeply grateful for your opinion.

We will work for better research.

Thank you.

Reviewer 2 Report

The manuscript entitled “Risk Factors on the Progression to Clinical Outcomes of COVID-19 patients in South Korea: Using National Data” by Kim et al., describes the risk factors associated with the progression to severity stages of COVID-19 in patients from South Korea. The study is very important for the better management of current pandemic when no vaccine or drug is available. Overall this is an interesting manuscript. However, some minor modifications are needed to further improve the manuscript.

English language needs to be improved as it is hard to understand the manuscript at multiple places. It has to be revised by a native speaker.

In abstract, the number of patients should be changed to 2,959 as this was the actual number used in the study.

Line 45: “Due to rapid increasing of infected patients, World Health Organization finally declared pandemic, which was as an international public health emergency [5,6].” This should be modified as a disease is pandemic when it spreads over entire country or globally.

Line 47: “In February, the country experienced an exponential spike in the number of COVID-19 cases following the emergence of a “super-spreader”[7]. As such, in South Korea, after the COVID-19  infection  response  guidelines  were  strengthened  and  after  the  infectious-disease  crisis  warning was raised to “severe” on February, the COVID-19 Code  of  Conduct  to  be  observed  at  the  severe disease stage was announced, and strong “social distancing” or physical distancing was implemented from March[8]. Not clear. Need to be modified.

Line 56: “Furthermore, any scientists of worldwide did not yet develop efficient medicines and vaccines for COVID-19 disease by misfortune. Therefore, identifying characteristics and risk factors of COVID-19 could provide valuable lesson.” Need to be modified for English language.

Figure 1 should be described properly in both figure legend as well as manuscript text. For example: it is not clear what represents blue or red lines. In Line 142, what is “every 10 years increase”? Text should be understandable.

Author Response

Open Review 2

The manuscript entitled “Risk Factors on the Progression to Clinical Outcomes of COVID-19 patients in South Korea: Using National Data” by Kim et al., describes the risk factors associated with the progression to severity stages of COVID-19 in patients from South Korea.

à The study is very important for the better management of current pandemic when no vaccine or drug is available. Overall this is an interesting manuscript. However, some minor modifications are needed to further improve the manuscript.

English language needs to be improved as it is hard to understand the manuscript at multiple places. It has to be revised by a native speaker.

In abstract, the number of patients should be changed to 2,959 as this was the actual number used in the study.

à We have modified it to your opinion.

Line 45: “Due to rapid increasing of infected patients, World Health Organization finally declared pandemic, which was as an international public health emergency [5,6].” This should be modified as a disease is pandemic when it spreads over entire country or globally.

à I changed into “As COVID-19  spreads over entire country or globally, World Health Organization finally declared pandemic [5,6].

Line 47: “In February, the country experienced an exponential spike in the number of COVID-19 cases following the emergence of a “super-spreader”[7]. As such, in South Korea, after the COVID-19 infection  response  guidelines  were  strengthened  and  after  the  infectious-disease  crisis  warning was raised to “severe” on February, the COVID-19 Code  of  Conduct  to  be  observed  at  the  severe disease stage was announced, and strong “social distancing” or physical distancing was implemented from March[8]. Not clear. Need to be modified.

à I changed into “ In February, the country experienced an exponential spike in the number of COVID-19 cases caused by a “super-spreader” [7]. After that time, in South Korea, the COVID-19 infection response guidelines were strengthened and the infectious-disease crisis warning was raised to “severe” on February. Also ‘the COVID-19 Code of Conduct’ to be observed at the severe disease stage was announced, and strong “social distancing” was implemented from March [8] “.

Line 56: “Furthermore, any scientists of worldwide did not yet develop efficient medicines and vaccines for COVID-19 disease by misfortune. Therefore, identifying characteristics and risk factors of COVID-19 could provide valuable lesson.” Need to be modified for English language.

à I changed to “Scientists all over the world are developing efficient medicines and vaccines for COVID-19 disease, but it will take a long time for it to be commercialized. Therefore, identifying characteristics and risk factors of COVID-19 could provide valuable lessons.”

Figure 1 should be described properly in both figure legend as well as manuscript text. For example: it is not clear what represents blue or red lines. In Line 142, what is “every 10 years increase”? Text should be understandable.

à We have modified it to your opinion.

We are deeply grateful for your opinion.

We will work for better research.

Thank you.

Reviewer 3 Report

Thank you very much for the manuscript. The topic is very interesting and relevant to provide information that supports clinical practice and appropriate patient management.

Below I make some comments that can help improve the presentation of the article:

Abstract

Line 16-18: establish more clearly the basis for studying progression of severity and severity of covid-19

Line 18: I would suggest write: “…5,628 confirmed COVID-19 patients were considered…” Or “…2,959 confirmed COVID-19 patients were analyzed using national data…

Line 22-23: Describe clearly that you considered two different outcomes (progression to severity stages of COVID-19 and severe Covid-19

Linea 24-28 Check redaction, it seems to be incomplete “ICU admission rate of older age, male, abnormal BMI, high heart rate,high body temperature, fever,cough,sputum, sore throat, rhinorrhea, fatigue, dyspnea, change of consciousness,diabetes mellitus, hypertension,chronic arterydisease, chronic kidney disease, cancer,dementia, abnormal hemoglobin, abnormal hematocrit, ,abnormal lymphocyte, abnormal platelets and abnormal white blood cell were high”

Introduction

Line 56-57: “Furthermore, any scientists of worldwide did not yet develop efficient medicines and vaccines for COVID-19 disease by misfortune”. I would suggest check redaction in order to recognize current advances and especially for vaccines, the speed at which studies are being developed

As in the abstract, it is not clear the importance of studying severity and progression of severity of covid-19.

Materials and Methods

Line 81: If possible, give reference of the data repository (website?)

Line 93-110: For the description of classifications and cut-off points considered in each variable (the criteria for considering normal and abnormal), try to put references that support the choice.

Better describe which are the main variables of analysis (dependent and independent variables) as well as which ones are considered as confounders.

Line 98-99: It is not clear which are the outcomes in this study. Here describe that the outcome was divided into death and release from isolation. Is it equivalent to talking about severity and progression of severity for covid-19?.

it is very important that you clearly describe the criteria considered to consider severe / non-severe case.

Statistical Analyses

The multivariate regression analysis used for progression of severity is not clear (linear regression?)

Check if the assumptions for this model are met, since, according to my understanding, the outcome is not numerical, but ordinal (8 levels?)

Results

Redaction of the presentation of all the results need to be checked. It is important not to repeat the same information that is already presented in the tables. Be more concrete.

Table 1: Check the % shown for age. Please present % of column as shown in the other variables.

Maybe if better to mention (next to the variable) when it is n and % or M and SD.

Please mention when t-test, Welch test, Cochran test and chi-square tests were used.

In order to simplify the table, maybe is better show only one category for variables with only 2 categories ((yes/no or Normal/abnormal), because it is understood that they are complementary and add up to 100%.

Multiple Logistic Regression for Risk Factors

it is not clear how age was entered into the model. From the description it seems that the age categories (every 10 years) were considered as numerical. In that case it doesn't seem appropriate to me.

Discussion

Discuss the limitations of the study and possible biases it might have. Also the possible effect of having complete data of 2959/5628, could it affect the results in something? (possibly worst record of mild cases)

Discuss the relationship and the concrete contribution of studying severity and progression of severity. In practice, what does it mean?

In the analysis was age explored as a modifier of the effect? (any interaction could be explored). Discuss this.

Author Response

Open Review 3

Abstract

Line 16-18: establish more clearly the basis for studying progression of severity and severity of covid-19

à I changed into “The identification of risk factors on progression to the severity stages is essential for providing more efficient and suitable management to COVID-19 patients.”

Line 18: I would suggest write: “…5,628 confirmed COVID-19 patients were considered…” Or “…2,959 confirmed COVID-19 patients were analyzed using national data…

à We have modified it to your opinion.

Line 22-23: Describe clearly that you considered two different outcomes (progression to severity stages of COVID-19 and severe Covid-19

à I added “ severe COVID-19”

Linea 24-28 Check redaction, it seems to be incomplete “ICU admission rate of older age, male, abnormal BMI, high heart rate,high body temperature, fever,cough,sputum, sore throat, rhinorrhea, fatigue, dyspnea, change of consciousness, diabetes mellitus, hypertension,chronic arterydisease, chronic kidney disease, cancer,dementia, abnormal hemoglobin, abnormal hematocrit, ,abnormal lymphocyte, abnormal platelets and abnormal white blood cell were high”

à I changed into “ICU (Intensive Care Unit) admission rate of patients having characteristics such as older age, male, abnormal BMI (Body Mass Index), high heart rate, high body temperature, fever, cough, sputum, sore throat, rhinorrhea, fatigue, dyspnea, change of consciousness, diabetes mellitus, hypertension, chronic artery disease, chronic kidney disease, cancer, dementia, abnormal hemoglobin, abnormal hematocrit, abnormal lymphocyte, abnormal platelets and abnormal white blood cell were high.”

Introduction

Line 56-57: “Furthermore, any scientists of worldwide did not yet develop efficient medicines and vaccines for COVID-19 disease by misfortune”. I would suggest check redaction in order to recognize current advances and especially for vaccines, the speed at which studies are being developed

à I changed into Scientists all over the world are developing efficient medicines and vaccines for COVID-19 disease, but it will take a long time for it to be commercialized.”

As in the abstract, it is not clear the importance of studying severity and progression of severity of covid-19.

à There is a content in the introduction. “The identification of risk factors on progression to the severity stages is essential for providing more efficient and suitable management to COVID-19 patients.”

Materials and Methods

Line 81: If possible, give reference of the data repository (website?)

à I addedhttp://www.kdca.go.kr/

Line 93-110: For the description of classifications and cut-off points considered in each variable (the criteria for considering normal and abnormal), try to put references that support the choice. Better describe which are the main variables of analysis (dependent and independent variables) as well as which ones are considered as confounders.

à We have modified it to your opinion.

Line 98-99: It is not clear which are the outcomes in this study. Here describe that the outcome was divided into death and release from isolation. Is it equivalent to talking about severity and progression of severity for covid-19?.

it is very important that you clearly describe the criteria considered to consider severe / non-severe case.

à I deleted “the outcome was divided into death and release from isolation”. And I added “The severe COVID-19 were classified into two groups such as severe case and non-severe case. Severe case included no disruption to daily life, hindrance to daily life and no oxygen required, oxygen treatment via nasal cannula and oxygen mask. Non-severe case included non-invasive ventilation, invasive ventilation, ECMO and death.”

Statistical Analyses

The multivariate regression analysis used for progression of severity is not clear (linear regression?)

Check if the assumptions for this model are met, since, according to my understanding, the outcome is not numerical, but ordinal (8 levels?)

à The severity is divided into 8 lesvels. Therefore, the Cochran test to check the trend of the variable with the rank was conducted, but only significant differences between the two variables were confirmed. However, we used these eight levels as a continuity variable to identify the factors affecting the severity, and I don't think there will be much trouble in interpreting them.

Results

Redaction of the presentation of all the results need to be checked. It is important not to repeat the same information that is already presented in the tables. Be more concrete.

à We have modified it to your opinion.

Table 1: Check the % shown for age. Please present % of column as shown in the other variables.

Maybe if better to mention (next to the variable) when it is n and % or M and SD.

Please mention when t-test, Welch test, Cochran test and chi-square tests were used.

à We have modified it to your opinion.

In order to simplify the table, maybe is better show only one category for variables with only 2 categories ((yes/no or Normal/abnormal), because it is understood that they are complementary and add up to 100%.

à We have modified it to your opinion.

Multiple Logistic Regression for Risk Factors

it is not clear how age was entered into the model. From the description it seems that the age categories (every 10 years) were considered as numerical. In that case it doesn't seem appropriate to me.

à The ages are divided into 9 groups. Therefore, as the number increases, the age increases, so it was analyzed as a continuity variable. These groups were provided on a limited basis by the Centers for Disease Control and Prevention. Therefore, we used it as a continuity variable to identify the influencing factor.

Discussion

Discuss the limitations of the study and possible biases it might have. Also the possible effect of having complete data of 2959/5628, could it affect the results in something? (possibly worst record of mild cases)

à Research limitations have been added.

Discuss the relationship and the concrete contribution of studying severity and progression of severity. In practice, what does it mean?

à I added contribution of studying severity and progression of severity.

In the analysis was age explored as a modifier of the effect? (any interaction could be explored). Discuss this.

à I added “If age was considered as a modifier, the result could be different. But this study focused on characteristics in a clinical situation, so the result revealed older age increased risk for progression to the severity stage.”

We are deeply grateful for your opinion.

We will work for better research.

Thank you.

Reviewer 4 Report

The authors discussed the risk Factors on the Progression to Clinical Outcomes
of COVID-19 patients in South Korea. I have few concerns regarding the current study:

  1. Overall the study written by very poor way, sounds not good in all sections either from the English and or scientific side.
  2. The authors did not provide any novel data and results through the whole study, I have a concern about conflict of interest related to the following published article. https://erj.ersjournals.com/content/early/2020/07/09/13993003.02144-2020
  3. The authors just repeat what have done before so far within the previous published data without adding any significant ideas
  4. All the manuscript sections were written very poor.
  5. Results did not have any novelty and not presented well
  6. Discussion is very poor

Author Response

Open Review 4

Overall the study written by very poor way, sounds not good in all sections either from the English and or scientific side.

à We have modified it as a whole.

  • The authors did not provide any novel data and results through the whole study, I have a concern about conflict of interest related to the following published article. https://erj.ersjournals.com/content/early/2020/07/09/13993003.02144-2020

à COVID-19 is currently in progress, and the reason it's difficult to develop vaccines is because the corona 19 virus becomes a strain. In this reality, data on confirmed cases in each country is very important, and this study was analyzed on basis of accurate data on all confirmed cases provided by the Korea Centers for Disease Control and Prevention. A paper published with data on confirmed cases provided by the Centers for Disease Control and Prevention is such an excellent study that it can be published in your journal without any present.

  • The authors just repeat what have done before so far within the previous published data without adding any significant ideas.

à The data provided by the Korea Centers for Disease Control and Prevention is very limited for the protection of patients' personal informations. Due to the limited information of the COVID-19 confirmed patients, the analytical methods that can be handled are limited. However, we took a step by step approach according to the severity progression of the patient, which can identify the risk factors according to the severity progression of the patient. So this study is different from other studies of COVID-19 confirmed cases focused on motality or severe stage. Therefore, it contains very important information that can provide effective treatments and managements for COVID-19 confirmed patients based on risk factors.

  • All the manuscript sections were written very poor.

à I edited many parts as much as I could. And I added insufficient explanation.

  • Results did not have any and not presented well

à I changed all result presentation.

  • Discussion is very poor

à We have modified it as a whole.

We are deeply grateful for your opinion.

We will work for better research.

Thank you.

Reviewer 5 Report

I read this paper with a lot of interest. It has an important message to deliver. The authors studied risk factors on the progression to severity stages of COVID-19 using few national data in South Korea. However, some points need to be improved, with the most important to analyze the error of statistical parameter estimation due to a few samples restriction.

  1. The COVID, ICU, etc. abbreviations are not fully given in the abstract. Before using an abbreviation, no matter how obvious it is, it should be defined in the text and abstract.
  2. Figure 1 needs editing. The vertical axis is unreadable.
  3. There are some errors in the typing, for example, online 27 it has two repeated commas, on lines 126 and 127 the numbers are smaller than the letters; review the entire manuscript and correct these errors.
  4. The main problem of the paper is the lack of error analysis. The classical statistic hypothesis is not satisfied. In the general case, the estimator of a random variable depends on sample size, distribution, and the degree of probabilistic dependence between samples. However, to improve the results is necessary to analyze the error introduced for sample size in the final result.

Author Response

Open Review 5

  • The COVID, ICU, etc. abbreviations are not fully given in the abstract. Before using an abbreviation, no matter how obvious it is, it should be defined in the text and abstract.

à We have modified it to your opinion.

  • Figure 1 needs editing. The vertical axis is unreadable.

à The vertical axis of the figure is the variable name on the left, and the OR value (confidence section) on the right. Therefore, it represents the OR value (right) according to the variable (left).

  • There are some errors in the typing, for example, online 27 it has two repeated commas, on lines 126 and 127 the numbers are smaller than the letters; review the entire manuscript and correct these errors.

à We have modified it to your opinion.

  • The main problem of the paper is the lack of error analysis. The classical statistic hypothesis is not satisfied. In the general case, the estimator of a random variable depends on sample size, distribution, and the degree of probabilistic dependence between samples. However, to improve the results is necessary to analyze the error introduced for sample size in the final result.

à  To consider sample size, distribution, and the degree of probabilistic dependence between samples, I used various analysis. If the assumption of equal variance for each variable was not met, welch's test (Bartlett test of homogeneity of variances; p<0.05) was used. Also, if the number of variables was less than 5, Fisher's exact test (N<5) was performed. The contents are commented under Table 1.

We are deeply grateful for your opinion.

We will work for better research.

Thank you.

Round 2

Reviewer 4 Report

None